# Educational Preparation and Course Approach of Undergraduate Sports Nutrition instructors in Large U.S. Institutions

**DOI:** 10.3390/sports11090176

**Published:** 2023-09-05

**Authors:** Kayla Marie Norton, Randall Spencer Davies, James Derek LeCheminant, Susan Fullmer

**Affiliations:** 1Department of Nutrition, Dietetics and Food Science, Brigham Young University, Provo, UT 84602, USA; 2Department of Instructional Psychology and Technology, Brigham Young University, Provo, UT 84602, USA

**Keywords:** sports nutrition education, higher education, sports nutrition curriculum, instructor preparation, nutritionist, exercise physiologist

## Abstract

College courses are often offered from various disciplines, and depending on which department offers the class, the course could be taught by faculty with different educational preparation or training. This could result in significant differences in the approach and content of the course (i.e., theoretical or applied) or a difference in the instructors’ perceived importance and, therefore, the depth and time spent on various topics. We evaluated potential differences in the sports nutrition curriculum because it is a course that is usually taught by either nutritionists or exercise physiologists. A cross-sectional survey was sent to sports nutrition instructors at accredited large U.S. institutions. Descriptive statistics were analyzed via an ANOVA and *Χ*^2^ using Crosstabs in Qualtrics. Alpha was set at *p* < 0.001. Additionally, short interviews with some participants were recorded and transcribed verbatim. The findings of this study indicated that regardless of the instructor’s educational preparation and discipline, the majority of sports nutrition topics received similar time and depth and were rated as similarly important (*p* > 0.001). Out of 10 current textbooks, the majority of instructors preferred only 1 of 4 of them. From the short interviews, instructors reported that their courses were more applied than theoretical or balanced between the two. Most instructors designed their courses with a focus on achieving applied outcomes.

## 1. Introduction

To maximize athletic performance, athletes should consume a balanced diet that matches their exercise regimen Ref. [1]. As such, they need access to appropriate nutrition recommendations. Evidence-based sports nutrition should integrate the science of sports nutrition with practical applied principles Ref. [2]. Providing students with knowledge about both the theory and application of nutrition principles prepares them to help athletes, or as an athlete, perform to the best of their abilities [1].

Sports nutrition is a science that has emerged over the last 40 years and provides accurate, evidence-based nutrition information to athletes, university instructors, and professionals who work with athletes. It is offered as an elective or required course for many majors, but the majority of students who enroll in a sports nutrition course are dietetic, nutritional science, exercise science, and athletic training majors. These students are most likely to work with athletes in their professional careers such as sports nutritionists, sports dietitians, athletic trainers, exercise physiologists, or sports medicine doctors, meaning that it is important that these students are adequately taught the correct scientific foundational knowledge and skills to work with athletes. They might work with all classes of athletes, from amateurs to Olympians or professionals. Knowing and teaching sound, evidenced-based sports nutrition principles will help maximize athletic performance while maintaining the athlete’s emotional and physical well-being. Additionally, other students might take the course for their own personal interests who also need to be taught evidenced-based nutritional concepts to support their physical activity while also maintaining emotional and physical well-being.

University courses are taught by faculty with diverse educational degrees, training, or experience in the subject matter. While some disciplines require specific faculty credentials, knowledge, and skills (e.g., nursing, engineering), many courses are taught by faculty who likely have related but maybe not specific education or training in the subject matter Refs. [3,4]. Based on the faculty member’s education and experience, the approach to a course could vary significantly. Currently, there are no published studies on how college courses are approached based on the instructor’s background and education.

The aims of the present study were to describe the various types of educational degrees and teaching experience of sports nutrition instructors and to determine if there is a difference between an instructor’s background with each of the following: textbook preference, the overall emphasis of course topics (applied vs. theoretical), and course content (depth, breadth, and time spent on various topics).

## 2. Materials and Methods

### 2.1. Survey Development

An online survey and short follow-up interview were developed with questions related to commonly taught sports nutrition topics. Survey topics were identified through sports nutrition textbooks [5,6,7,8,9,10,11,12,13,14] and with the help of a dissertation Ref. [2]. The survey questions asked instructors to estimate how much total time they spent on each topic throughout the term/semester within 5 min increments (1–5, 6–10, 11–15, 16–20, or >20 min). Instructors were also asked to rate the depth and perceived importance of each topic on a scale of 1–10. The specific topics can be found in the tables, including Appendix A. A copy of the survey is included as Appendix A.

The survey included three sections: instructor demographics (12 questions), course demographics (15 questions), and an evaluation of course content (21 questions). The research team, consisting of a registered dietitian nutritionist (RDN), exercise scientist, and evaluator of educational curriculum, reviewed the survey for clarity and completeness. Pilot studies were conducted to review the survey and short interview with current or previous sports nutrition instructors. The participants provided verbal input while answering each question. As a result of the pilot interviews, the survey had only minor modifications to improve flow, participant understanding, and decrease survey fatigue. The survey included 45 total questions, and the short interview questions were narrowed down to 7.

### 2.2. Participants

The six accreditation bodies in the Council for Higher Education were contacted (Higher Learning Commission, Middle States Commission on Higher Education, New England Commission of Higher Education, Northwest Commission on Colleges and Universities, Southern Association of Colleges and Schools Commission on Colleges, and WASC Senior College and University Commission) to obtain lists of all large (defined as more than 10,000 enrolled students) four-year institutions. Once the lists were obtained, institutions were excluded if they did not offer an undergraduate sports nutrition course. A total of 324 institutions were identified. Research assistants then went to each university’s website to identify if a sports nutrition course was offered. There were 189 universities that offered a sports nutrition course, with 217 instructors. Emails were identified from the websites of each institution.

### 2.3. Recruitment

Potential participants were sent an email containing an overview of the study along with an informed consent form. If the instructor was willing to participate, they followed a link to a Qualtrics survey Ref. [15]. The final question of the survey asked if they would be willing to take part in a short follow-up interview.

### 2.4. Institutional Review Board Approval

All study procedures were approved by the university’s Institutional Review Board (#IRB2021-296), and all participants provided informed consent for both the survey and the short interviews.

### 2.5. Data Analysis

Participants were categorized based on the following criteria: degree type (nutrition-only, exercise-physiology-only, or both nutrition and exercise physiology degrees), age (<35 years old, 35–44 years old, and >45 years old), and experience, the number of times they had taught the course (novice [<3 times], experienced [3–6 times], and veteran [>6 times]).

Data were analyzed using Crosstabs in Qualtrics [15]. A one-way analysis of variance (ANOVA) and CHI square tests were conducted within each of the three main categories (degree type, age, experience) and the topics identified in the course content section of the survey. Regarding the depth and importance of each topic, mean scores on a scale of 1–10 were collapsed as follows: 1–3 = least depth/importance, 3.1–6.9 = moderate depth/importance, and 7.0–10 = most depth/importance. To reduce the risk of a type 1 error and to account for multiple comparisons, statistical significance was set at an alpha of *p* < 0.001. After conducting the analysis, there were some statistically significant differences in the same mean scale groupings (i.e., 7 was significantly less than 9 but both 7 and 9 were in the “most” category), but these were not considered practically significant.

All of the interviews were conducted one-on-one via Zoom by a single researcher (SF) who is an RDN and has experience conducting short interviews and extensive teaching experience with the course. Prior to the study, the interviewer did not know any of the subjects. At the beginning of the interview, the facilitator explained the purpose/reason for the study and oral consent was obtained. All the participants were asked the same seven questions. The interviews lasted 5–10 min. There were no repeat interviews. The intent of these follow-up questions was to allow participants to clarify any of their responses to the survey and answer the seven questions. The subjects received a USD 25 Amazon gift card at the end of the short interview.

Interviews were transcribed verbatim by two trained research assistants. Once the interviews were transcribed, an independent content analysis was performed by each research assistant. Each read the responses to the seven questions and categorized them into common themes, then the assistants compared their analysis with one another. Most answers were concise and similar across respondents; therefore, common themes were easy to identify. Agreement was 100% on the first round [16].

## 3. Results

### 3.1. Survey Response

Of the 324 institutions that were identified, 119 schools did not have or were not currently teaching an undergraduate sports nutrition course, and 16 universities did not have instructor contact information. Overall, 189 qualifying institutions were identified, and 217 instructors were sent a recruitment email. Nine emails were returned with invalid email addresses. Of the 208 emailed surveys, 65 instructors responded; 10 were excluded due to incomplete survey responses, and 13 were excluded because they only taught the course online. Only one instructor did not have a degree in nutrition or physiology, so it was decided to exclude that instructor from the analysis so the comparison between nutrition, physiology, or a combination of both degrees could be evaluated. In total, 41 surveys were included in the analysis (20% response rate).

Of the potential participants who did not respond to the survey (*n* = 153), 56 (42%) held nutrition-related degrees, 49 (34%) held exercise-physiology-related degrees, 19 (14%) held one of each degree, 8 (6%) did not have a nutrition- or exercise-physiology-related degree, and 20 (12%) were unable to be identified.

Of the participants that were excluded from the survey (*n* = 21), 11 (46%) held nutrition-related degrees, 4 (17%) held exercise-physiology-related degrees, 4 (17%) held one of each degree, 1 (4%) did not have a nutrition- or exercise-physiology-related degree, and 4 (17%) were unable to be identified. Therefore, we believe the included respondents are representative of the broader population of sports nutrition instructors.

In total, 27 instructors agreed to participate in the short interview, but only 22 participants completed it. Also, 1 interview failed to record, leaving 21 usable interviews (51% interview response rate).

### 3.2. Instructor Demographics

Within the Degree category, 18 instructors had nutrition-only degrees (NODs), 12 had exercise-physiology-only degrees (EPDs), and 11 instructors had both nutrition and exercise physiology degrees (BNE). Within the Age category, 10 instructors were under 35 years old (young instructor or YI), 14 were between 35 and 44 years old (middled-age instructor or MI), and 17 were older than 45 years old (older instructor or OI). Most of the results based on instructor’s age paralleled the number of times they had taught a course, so results based on age are omitted from the remainder of this paper. Within the Experience category, 6 instructors had taught the course less than three times (novice), 10 had taught the course three to six times (experienced), and 25 had taught the course more than six times (veteran). Eleven instructors were master’s prepared, six had doctoral degrees but no master’s degree, twenty-three had both master’s and doctoral degrees, and one was bachelor’s prepared.

Almost half (46%) of the instructors were registered dietitians. Other credentials held by the instructors were as follows (in descending order): twelve had “other credentials”, six were certified specialists in sports dietetics, four were certified strength and conditioning trainers, two were American College of Sports Medicine-certified exercise physiologists, two were American Council of Exercise-certified, and one was a certified nutrition specialist. These credentials were not exclusive, meaning that the instructors could have held more than one or had no credentials.

### 3.3. Course Demographics

In total, 61% of sports nutrition courses were taught face-to-face (*n* = 25), 20% were taught as hybrid (i.e., both in-person and online) (*n* = 8), and the others were taught in person or online depending on the semester and COVID-19 implications (*n* = 8). Also, 41% of courses were taught in a nutrition (*n* = 17) department and 29% were in a combined nutrition and exercise science department or cross-listed in both (*n* = 12). Nine were taught in exercise science departments and one each was taught in public health, biomedical sciences, and family and consumer sciences.

Eighty-eight percent of courses were worth three credits (*n* = 36) and taught for over 15 weeks in a semester (*n* = 27). Most courses were junior (*n* = 14) or senior (*n* = 21) level, three were taught at a sophomore level, and three were graduate level that allowed undergraduate students. The most common pre-requisite course was basic nutrition (*n* = 28), followed by human physiology (*n* = 17) and exercise physiology (*n* = 7).

Twenty-seven instructors (66%) required a textbook and viewed the textbook as being important to their class (mean 7.3 out of 10). Common reasons for textbook preference reported in the interview included recommendations by other instructors, textbooks updated with new research, good application of the science, cost, or adequate levels of biochemistry and physiology. In total, 35 instructors (83%) used supplemental readings, 13 of which did not require a textbook for their course. Of the instructors that required a textbook, 22 (77%) also used supplemental readings. Three (7%) did not require a textbook or use supplemental readings. The instructors who used supplemental readings rated them in similar importance to the textbook (mean 7.6 out of 10).

The majority of instructors (74%) used 1 of 4 sports nutrition textbooks [8,9,10,14]. Seven other textbooks were used by one instructor each [5,6,7,11,12,13,17], and the rest of the instructors did not require a textbook. There were no significant differences in educational background and choice of textbook, though instructors with at least one nutrition degree tended to use 1 of 3 specific textbooks [9,10,14], while EPD holders tended to use the textbook authored by Jeukendrup [8].

### 3.4. Course Content

#### 3.4.1. Carbohydrates

On average, instructors focused the most time, depth, and importance within the subject of carbohydrates, teaching about the timing of recommendations, utilization, and estimating carbohydrate needs (Table 1). Comparing instructors by their degree type, 64% of BNE holders, 42% of EPD holders, and 59% of NOD holders reported spending more than 20 min discussing the timing of carbohydrate intake (NS). In total, 67% of novice instructors rated the importance of dietary sources as a six or lower, while 60% of experienced and 80% of veterans rated it as a seven or higher (NS)). In contrast, instructors spent the least amount of time, depth, and importance teaching about structures, writing sample diets, and the digestion and absorption of carbohydrates (Table 1).

#### 3.4.2. Proteins

The most time, depth, and importance in terms of protein was spent on estimating protein needs, timing of protein recommendations, and protein utilization (Table 2). The majority of BNE (82%) and NOD (56%) holders reported spending more than 20 min on estimating needs, and 83% of EPD holders were split evenly between 16 and 20 or more than 20 min (NS). Regarding how deep instructors taught the timing of protein recommendations, 55% of BNE and 61% of NOD holders ranked it as a 10, while only 25% of instructors with an EPD ranked it as a 10 (NS). Estimating protein needs was considered an important topic, as 50% of novice and 48% of veteran instructors ranked it as a 10, and 80% of experienced instructors ranked it as an 8 or higher (NS). The least amount of time, depth, and importance for protein topics was focused on protein structures, writing sample diets for protein, and protein storage (Table 2).

#### 3.4.3. Fats

Fat utilization, metabolic pathways, and estimating fat needs received the most time, depth, and importance within the fat subjects (Appendix A). Almost half of BNE (46%) and NOD (44%) holders spent more than 20 min on fat utilization, while 42% of those with an EPD spent 16–20 min (NS). Looking at the depth of fat utilization, 50% of EPD holders ranked it as a 9 compared with NOD holders, in which 50% ranked it as a 10 (NS. With respect to the importance of dietary fat sources, 33% of novice instructors ranked it as only as a two compared to experienced and veterans, whose responses varied. In total, 30% of experienced instructors ranked it as a four and 20% ranked it as a nine, and 68% of veterans ranked it between five and eight (NS). The least amount of time, depth, and importance spent on fat topics was on fat loading and fat structures (Appendix A).

#### 3.4.4. Fluids

Fluid topics received the most class time, depth, and importance on average compared to the individual macronutrients. Specifically, the timing of fluid intake, over/under hydration, and assessing fluid needs were focused on the most (Table 3). There were no ratings below a five for how deep instructors taught the timing of fluid recommendations, and regardless of degree, age, or experience, 71% of instructors rated this topic as an eight or higher (NS). When considering the importance of over- and under-hydration, 30% of BNE holders ranked it as a 7 compared to 67% of EPD holders, who ranked it as an 8 or 9, and 56% of NOD holders, who ranked it as a 10 (NS). The fluid topics that received the least class focus were fluid absorption, environmental factors affecting fluid status, and fluid functions (Table 3).

#### 3.4.5. Body Composition and Weight Management

On average, instructors focused the most time, depth, and importance within this topic on determining energy needs and methods of determining a healthy body weight, and the least amount was spent on sport-specific percent body fat and percent body fat in males and females (Appendix A). Regardless of degree, age, or experience, over 67% of instructors spent more than 20 min on estimating energy needs. In terms of how deep instructors taught body composition goals for athletes, 100% of novice instructors ranked it as an eight or higher, and the majority of experienced and veterans ranked it as a seven or lower. The majority of novice (60%) and experienced instructors (60%) rated the importance of methods of determining a healthy body weight as an eight or higher, while 54% of veterans ranked it as a seven or lower (NS).

#### 3.4.6. Eating Disorders

Instructors focused the most time, depth, and importance within the subject of eating disorders on the Female Athlete Triad, disordered eating, and signs and symptoms of eating disorders, and the least amount of time, depth, and importance was spent on orthorexia, hormonal adaptations, and treatments (Table 4). Based on instructor experience, half of the novice instructors spent more than 20 min on signs and symptoms, which was more than 80% of experienced instructors who spent less than 10 min (NS). In terms of how deep instructors taught signs and symptoms, 60% of novice instructors ranked it as a 10, 63% of experienced ranked is as a six or seven, and veterans were split, with 45% having ranked it as a five or less and 55% ranked it as a seven or higher (NS). Seventy-one percent of all instructors, regardless of degree, age, or experience, ranked the Female Athlete Triad as a seven or higher in terms of importance to their course.

#### 3.4.7. Vitamins and Minerals

On average, instructors spent the most time discussing calcium, vitamin D, iron, sodium, and potassium and the least amount of time on vitamin K (Appendix A). When looking at time spent based on degree type, 60% of BNE holders were evenly split between 11 and 15 and 16 and 20 min on iron, while 33% of EPD holders spent 1–5 min, and the responses of those with NODs were varied (NS).

#### 3.4.8. Ergogenic Aids

The three most commonly discussed ergogenic aids were whey protein, creatine, and caffeine. On average, instructors spent the least amount of time on pyruvate, blood doping, and androstenedione (Appendix A). Specifically, 73% of EPD holders spent 11–15 min on caffeine compared to 55% of BNE holders and 50% of NOD holders, who spent less than 10 min (NS). In total, 92% of EPD holders spent less than 10 min on carnitine, 39% of NOD holders spent 11–15 min, and 64% of BNE holders spent 1–5 min (NS).

#### 3.4.9. Short Interviews

Of the 21 instructors that participated in the short interviews, 15 reported that their courses were more applied than theoretical, 4 felt their course was a balance of application and theory, and 2 believed their course leaned towards theory. Some of the application-based activities included case studies (*n* = 5), student research projects (*n* = 5), and personal application of the material (*n* = 3). Others that were reported by only one instructor each included working with athletes, diet planning exercises, identifying different dietary patterns, and using a lab to make healthy recipes.

Instructors stated that the biggest influence on their approach to the course came from student interest (*n* = 13), the instructor’s educational background (*n* = 10), working with athletes (*n* = 7), or personal experience as an athlete (*n* = 5). When asked about how their teaching has changed since the time they began teaching the course, instructors reported making the class more applied (*n* = 4), including more student interaction (*n* = 3), or student and instructor feedback (*n* = 2). Two instructors indicated that their courses were constantly changing. To stay current in their fields, instructors attend various conferences, including those from the American College of Sports Medicine, Sports and Human Performance, Collegiate and Professional Sports Dietitian’s Association, Sports, Cardiovascular and Wellness Nutrition, American Society for Nutrition, Food and Nutrition Conference and Expo, International Society for Clinical Densitometry, and Gatorade. Ten of the instructors conducted sports-nutrition-related research, and five tried to stay current by reading the literature.

## 4. Discussion

The purpose of this study was to evaluate higher education sports nutrition courses to identify the types of academic credentials of instructors and then to determine how course content, defined as depth, breadth, importance, and overall emphasis of course topics (applied vs. theoretical), varies based on the educational background and experience of the instructor. Our primary finding was that, regardless of educational background or teaching experience, there were minimal to no practical differences in how instructors spent class time on various topics. Additionally, none of the topics were considered to be of least depth or importance (average of 1–3.0), likely suggesting that the topics covered in textbooks and course content are truly important for sports nutrition courses. This was an unexpected finding. We hypothesized a priori that there would be differences in the time spent on applied and theoretical concepts based on the instructor’s background. However, from conducting a comparison of 10 sports nutrition textbooks used as resources to design the survey, we observed extensive consistencies in the topics covered (chapters) and content in each chapter. As reported in the short interviews, faculty who teach sports nutrition courses have memberships with professional sports nutrition organizations and practice groups (e.g., Academy of Nutrition and Dietetics, American College of Sports Nutrition) and conduct research in the field. As they read relevant journals and attend professional meetings, the presentation of research and recommendations is consistent across disciplines. This suggests that the science of sports nutrition is a cohesive discipline [18,19,20,21,22,23].

There were essentially two types of educational backgrounds in our sample: physiology and nutrition. Other than one respondent who did not have a background in either area and was therefore excluded from analysis, each participant in our sample had at least one expected educational background. Additionally, both educational backgrounds were sufficiently represented in the analysis, and therefore we are confident the sample size was adequate to represent the characteristics of our primary population Ref. [24]. An analysis of all the eligible universities identified for this study (nonrespondent, excluded responses and included subjects) found that over 85% of sports nutrition courses at large US Universities are taught by instructors with degrees in nutrition and or exercise physiology.

Textbooks account for a large portion of student expenses Refs. [25,26]. A theme identified in the short interviews was that instructors try to find ways to decrease the burden of textbook costs. In total, 66% of instructors used a textbook for the course; however, the majority of instructors (83%) used supplemental readings, either exclusively or in conjunction with the textbook. Studies show that supplemental reading provides critical reading and thinking skills for both instructors and students [27], which is likely one of the reasons why so many instructors include them in their course. Supplemental readings are also a way for students to learn the course topics and save money.

Lastly, we wanted to determine if an instructor’s background influenced the overall course approach as either applied (how and what an athlete should eat) or theoretical (focus on metabolism, functions of nutrients) and to determine if there is a significant difference in the breadth and depth of the core sports nutrition topics. Our survey asked about the timing, depth, and importance of both theoretical (structures, functions, digestion and absorption, metabolic pathways, utilization, and storage) and applied (estimating dietary needs, dietary sources, practicing sample diets, and timing of recommendations) topics of major subjects typically taught within a sports nutrition courses [2].

About 90% of the instructors who participated in the short interview reported that their course was more application-based or a balance of applied and theoretical topics. This was confirmed by the observation that many of the topics that received the most time, depth, and importance from the survey were applied concepts. However, no differences were seen in the approach to the course when compared with the instructors’ educational backgrounds.

Concepts that are difficult to explain would naturally take more time than easier concepts Ref. [28]. For example, instructors reported spending more time on the metabolic pathways of macronutrients and less time on topics like macronutrient structures. Instructors reported the most depth and importance for those topics that also received the most class time.

### 4.1. Carbohydrates

Carbohydrates received a lot of lecture time regardless of the instructor’s educational background. The theorical concepts of function, metabolic pathways, utilization, and storage were each given 16–20 min of class time (Table 1) and, except for functions, all were considered “most” important and had the “most” depth. For each of these topics, over half of instructors spent a minimum of 16 min teaching these concepts. More than 50% of both EPD and NOD instructors spent over 20 min teaching students how to estimate the carbohydrate needs of athletes, and over 50% spent at least 16 min teaching the dietary sources of carbohydrates. Overall, there was a similar and generous amount of time spent teaching theoretical and applied carbohydrate principles by both instructor types. A combination of applied skills and a scientific foundation enables students to personalize recommendations for athletes.

Adequate carbohydrates have long been a backbone to athletic success. Both high- and low-carbohydrate intake strategies play an important role in providing high carbohydrate availability Ref. [23]. However, in general, most athletes benefit from consuming a high-carbohydrate diet before, during, and after exercise Ref. [23]. Therefore, it is not surprising that of the 10 subjects in carbohydrate lectures, both instructor groups rated 6 of the 10 carbohydrate concepts as “most” important.

### 4.2. Proteins

Not surprisingly, both groups of instructors rated protein function, utilization, determining protein needs, and the timing of protein intake as “most” important, and each of these topics were discussed for 16–20 min. Similarly, estimating needs, identifying dietary sources, and the timing of protein intake were discussed for 16–20 min (Table 2). About 40% of both EPD and NOD instructors spent over 20 min discussing functions of protein. Interestingly, 82% of EPD instructors spent over 20 min on how to estimate protein needs compared to 42% of NOD instructors. Likewise, 45% compared to 17% of EPD and NOD instructors, respectively, spent over 20 min discussing dietary sources of protein). Consuming adequate protein is necessary for tissue repair, anabolism, and other metabolic adaptations with exercise and training [29,30,31].

### 4.3. Fats

Theoretical-based principles like fat metabolism and utilization received the most time and were rated as “most” importance and to receive the most depth by both groups, likely because these concepts take time to explain. Functions, digestion/absorption, storage, and estimating the dietary needs of fat received 11–15 min of time by each group (Appendix A). NOD instructors tended to spend more time on dietary sources and the timing of recommendations compared to EPD instructors within the fat topics (Appendix A). However, neither group spent more than 10 min on fat loading, and 27% of instructors did not even teach it in their course.

Fatty acids are a major fuel source for ATP generation, especially during light-to-moderate exercise and endurance events Ref. [5]. Fatty acids can generate more ATP per molecule than glucose, though it is less efficient and requires more oxygen Refs. [32,33]. Ultimately, instructors reported that the most time, depth, and importance for fat topics were primarily theoretical, including fat metabolism and utilization, though they also focused on the applied topic of estimating fat needs.

### 4.4. Fluids

Fluids were rated as the most important subject overall, and the fluid topics that received the most class time, depth, and importance were all applied topics, including timing of fluid intake, problems associated with over- or under-hydration, and assessing fluid needs (Table 3). There were minimal differences regarding timing, depth, or importance between EPD and NOD instructors. With the exception of EPD holders and time spent on the functions of fluid, over half of all instructors spent at least 11–15 min on each concept identified within the fluid topics. And with a few exceptions, most instructors rated each topic as “most” important and rated the overall topic of fluids to receive the “most” depth. This is not surprising because fluid status can be greatly affected by exercise, and significant fluid deficits, known as hypohydration, can lead to compromised cognitive function, increased glycogen use, and impaired performance [34,35]. Overhydration can also be dangerous, resulting in lower plasma sodium levels, a complication known as hyponatremia [36,37].

### 4.5. Body Composition and Weight Management

Instructors reported spending the most time, depth, and importance in their courses on applied concepts, including determining energy needs and a healthy body weight (Appendix A). Body composition and body weight are important for many athletes, and athletes most vulnerable to unhealthy behaviors and weights are wrestlers, gymnasts, and endurance athletes [38], where low fat and body mass are desirable Ref. [23]. Providing athletes of all types and levels of expertise adequate energy to support their physical activity and training levels is a basic nutritional need and important to an athlete’s success. All groups reported spending 11–15 min on body composition goals for athletes and >15 min teaching students how to calculate energy needs (Appendix A).

### 4.6. Eating Disorders

Instructors reported spending the least amount of time, depth, and importance on the subject of eating disorders. All groups primarily rated the Female Athlete Triad “most” importance and depth (Table 4). When comparing time spent (NOD vs. EDP) in the various topics of eating disorders, a higher percentage of NOD instructors tended to spend more time on the types of eating disorders (39 vs. 17%), disordered eating (39 vs. 25%), signs and symptoms (44 vs. 8%), and treatments (28 vs. 17%). Instructors with nutrition training are more likely to have more foundation in eating disorders and, therefore, might feel more comfortable discussing these topics. While eating disorders require qualified professional intervention, it is important for students to recognize the signs and symptoms and to understand some of the factors that can contribute to disordered eating behaviors and eating disorders.

### 4.7. Vitamins and Minerals

To reduce participant fatigue, subjects were asked to only estimate the amount of class time they spent on vitamins, minerals, and ergogenic acids. Depth and importance were not assessed. The vitamins and minerals that instructors spent the most time on were vitamin D, calcium, sodium, potassium, and iron. Each of these topics were discussed for 11–15 min. All of the other nutrients we included in the survey had at least 6–10 min of discussion (Appendix A).

Calcium and vitamin D play key roles in bone health Ref. [39]. Their roles in bone metabolism and discussion on food sources take time to explain, so it is not surprising that these two nutrients each received longer than 5 min of class time (Appendix A).

Sodium and potassium are important electrolytes that are lost in sweat and play a large role in fluid and electrolyte balance Ref. [40]. Hyponatremia, usually caused by overhydration, results in low plasma sodium levels (<135 mmol/L) Ref. [41]. Incidences of hyponatremia in athletes are more common in endurance events lasting more than four hours or in less-trained individuals [36].

Iron is a key nutrient responsible for transporting oxygen throughout the body and is a component of the electron transport chain [23,42]. Some athletes are at an increased risk of iron deficiency due to decreased dietary intake, increased GI losses in some sports, inflammation from frequent exercise, and elevated hepcidin, resulting in decreased iron absorption [42,43].

### 4.8. Ergogenic Aids

The following ergogenic aids were discussed for 11–15 min by both groups: caffeine, creatine, leucine, branched chain amino acids, and whey protein (Appendix A). Caffeine has been extensively researched and is widely used by many athletes to enhance fatty acid oxidation and decrease the perceptions of fatigue and pain during exercise Refs. [44,45]. Creatine, caffeine, and whey protein supplements have been reported as the most commonly consumed supplements by athletes Refs. [45,46]. Creatine is a popular supplement in the literature due to its benefits for short-duration, high-intensity events like sprinting or weightlifting. Creatine specifically works by increasing intramuscular phosphocreatine stores, which are utilized to quickly replenish ATP at the onset of exercise Ref. [47]. In terms of whey protein, studies show that protein intake following resistance exercise results in increased muscle protein synthesis (MPS) Ref. [48]. Whey protein is quickly absorbed and supplies the body with the essential amino acids necessary for MPS and sports performance [23,48].

Of the ergogenic aids we included in the survey, glycerol, HMB, blood doping, androstenedione, DHEA, pyruvate, and ephedrine were only discussed for 1–5 min (Appendix A). It was surprising that anabolic steroids only received 6–10 min of time. However, androstenedione and DHEA were listed separately from steroids on the survey.

The primary limitation of this study was the use of self-reported data. Instructors were asked to estimate how much time they spent on various sports nutrition topics. Topics might be covered in multiple chapters, making it difficult to estimate the total time spent on a topic. The time spent on these topics likely varies between semesters depending on the students and the needs of the class. Additionally, the length of the survey might have contributed to survey fatigue, resulting in less accurate estimations. We believe the sample size is adequate and was representative of non-respondent characteristics. *p*-values were set at *p* < 0.001 to account for multiple statistical comparisons. For this reason, some significant differences that might have been present were not reported on because they did not meet the a priori *p*-value criteria. However, many of the results that we found suggested consistency despite the differing education and experience of instructors. Future research might examine how student preparedness to work with athletes may differ based on the various backgrounds or teaching approaches of their instructors.

## 5. Conclusions

The results of this study suggest that, regardless of a sports nutrition instructor’s educational or professional background, there are minimal or no differences in the amount of time, depth, or importance spent teaching the main sports nutrition topics in university settings. The common topics are covered by both instructor types and receive a similar amount of time and importance. The majority of instructors reported that their course was more applied- than theory-based; however, all courses had a balance of both. The majority of sports nutrition courses taught in large US institutions are by instructors with degrees in exercise physiology and/or nutrition. Practice implications of this research could be valuable for sports nutrition instructors to serve as a benchmark regarding how much time, depth, and importance each topic could be given in the course.

## Figures and Tables

**Table 1 sports-11-00176-t001:** **Comparison of the average time spent, depth, and importance of the major topics for carbohydrates reported by** sports nutrition instructors in a given semester based on the instructor’s degree, age, and experience teaching the course (N = 41).

	Instructor’s Educational Preparation			Number of Times Course Taught		
Topic	Both Nutrition and Exercise Physiology	Exercise Physiology Only	Nutrition Only	Novice (<3)	Experienced (3–6)	Veteran (>6)
**Structures**						
Time spent	6–10 min	6–10 min	6–10 min	6–10 min	6–10 min	6–10 min
Depth †	Moderate	Moderate	Moderate	Moderate	Moderate	Moderate
Importance ‡	Moderate	Moderate	Moderate	Moderate	Moderate	Moderate
**Functions**						
Time spent	16–20 min	16–20 min	16–20 min	16–20 min	16–20 min	16–20 min
Depth	Moderate	Moderate	Moderate	Moderate	Moderate	Moderate
Importance	Most	Moderate	Most	Most	Most	Moderate
**Digestion and Absorption**						
Time spent	11–15 min	16–20 min	11–15 min	11–15 min	11–15 min	11–15 min
Depth	Moderate	Moderate	Moderate	Moderate	Moderate	Moderate
Importance	Moderate	Most	Most	Most	Most	Moderate
**Metabolic Pathways**						
Time spent	16–20 min	16–20 min	16–20 min	>20 min	16–20 min	16–20 min
Depth	Most	Most	Most	Most	Most	Most
Importance	Moderate	Most	Most	Most	Most	Most
**Utilization**						
Time spent	16–20 min	16–20 min	16–20 min	>20 min	16–20 min	16–20 min
Depth	Most	Most	Most	Most	Most	Most
Importance	Most	Most	Most	Most	Most	Most
**Storage**						
Time spent	16–20 min	16–20 min	16–20 min	16–20 min	16–20 min	16–20 min
Depth	Most	Most	Most	Most	Most	Moderate
Importance	Most	Most	Most	Most	Most	Most
**Estimating Needs**						
Time spent	16–20 min	16–20 min	16–20 min	16–20 min	16–20 min	16–20 min
Depth	Most	Most	Most	Most	Most	Most
Importance	Most	Most	Most	Most	Most	Most
**Dietary Sources**						
Time spent	16–20 min	11–15 min	11–15 min	11–15 min	11–15 min	16–20 min
Depth	Moderate	Moderate	Moderate	Moderate	Moderate	Moderate
Importance	Most	Moderate	Most	Moderate	Most	Most
**Sample Diets**						
Time spent	16–20 min	6–10 min	11–15 min	11–15 min	11–15 min	11–15 min
Depth	Moderate	Moderate	Moderate	Moderate	Moderate	Moderate
Importance	Most	Moderate	Moderate	Moderate	Most	Moderate
**Timing of Recommendations**						
Time spent	16–20 min	16–20 min	16–20 min	16–20 min	16–20 min	16–20 min
Depth	Most	Moderate	Most	Most	Most	Most
Importance	Most	Most	Most	Most	Most	Most
**Carbohydrate Loading**						
Time spent	16–20 min	11–15 min	11–15 min	16–20 min	11–15 min	11–15 min
Depth	Most	Most	Most	Most	Most	Moderate
Importance	Most	Most	Most	Most	Most	Most

No significant different between groups. † Categories of Depth scores based on a scale of 1–10. Least: 1–3.0. Moderate: 3.1–6.9. Most: 7.0–10. ‡ Categories of Importance scores based on a scale of 1–10. Least: 1–3.0. Moderate: 3.1–6.9. Most: 7.0–10.

**Table 2 sports-11-00176-t002:** Comparison of the average time spent, depth, and importance of the major topics for **proteins** reported by sports nutrition instructors in a given semester based on the instructor’s degree, age, and experience teaching the course (N = 41).

	Instructor’s Educational Background			Number of Times Course Taught		
Topic	Both Nutrition and Exercise Physiology	Exercise Physiology Only	Nutrition Only	Novice (<3)	Experienced (3–6)	Veteran (>6)
**Structures**						
Time spent	6–10 min	11–15 min	11–15 min	11–15 min	6–10 min	11–15 min
Depth †	Moderate	Moderate	Moderate	Moderate	Moderate	Moderate
Importance ‡	Moderate	Moderate	Moderate	Moderate	Moderate	Moderate
**Functions**						
Time spent	16–20 min	16–20 min	16–20 min	16–20 min	16–20 min	16–20 min
Depth	Most	Most	Most	Most	Most	Most
Importance	Most	Most	Most	Most	Most	Most
**Digestion and Absorption**						
Time spent	11–15 min	11–15 min	11–15 min	16–20 min	11–15 min	11–15 min
Depth	Moderate	Most	Moderate	Most	Most	Moderate
Importance	Moderate	Most	Moderate	Most	Most	Moderate
**Metabolic Pathways**						
Time spent	11–15 min	16–20 min	16–20 min	>20 min	11–15 min	11–15 min
Depth	Moderate	Most	Most	Most	Most	Most
Importance	Moderate	Most	Most	Most	Most	Moderate
**Utilization**						
Time spent	16–20 min	16–20 min	16–20 min	>20 min	16–20 min	16–20 min
Depth	Most	Most	Most	Most	Most	Most
Importance	Most	Most	Most	Most	Most	Most
**Storage**						
Time spent	11–15 min	16–20 min	11–15 min	11–15 min	11–15 min	11–15 min
Depth	Moderate	Moderate	Moderate	Most	Moderate	Moderate
Importance	Moderate	Most	Moderate	Moderate	Most	Moderate
**Estimating needs**						
Time spent	>20 min	16–20 min	16–20 min	>20 min	16–20 min	>20 min
Depth	Most	Most	Most	Most	Most	Most
Importance	Most	Most	Most	Most	Most	Most
**Dietary sources**						
Time spent	16–20 min	16–20 min	16–20 min	11–15 min	16–20 min	16–20 min
Depth	Moderate	Most	Most	Most	Most	Most
Importance	Most	Moderate	Most	Moderate	Most	Most
**Sample diets**						
Time spent	11–15 min	6–10 min	11–15 min	6–10 min	11–15 min	11–15 min
Depth	Moderate	Moderate	Most	Moderate	Most	Moderate
Importance	Most	Moderate	Moderate	Moderate	Most	Moderate
**Timing of Recommendations**						
Time spent	16–20 min	16–20 min	>20 min	>20 min	16–20 min	16–20 min
Depth	Most	Most	Most	Most	Most	Most
Importance	Most	Most	Most	Most	Most	Most

No significant differences between groups. † Categories of Depth scores based on a scale of 1–10. Least: 1–3.0. Moderate: 3.1–6.9. Most: 7.0–10. ‡ Categories of Importance scores based on a scale of 1–10. Least: 1–3.0. Moderate: 3.1–6.9. Most: 7.0–10.

**Table 3 sports-11-00176-t003:** Comparison of average time spent, depth, and importance of the major topics for **fluids** reported by sports nutrition instructors in a given semester based on the instructor’s degree type, age, and experience teaching the course (N = 41).

	Instructor’s Education Background			Number of Times Course Taught		
Topic	Both Nutrition and Exercise Physiology	Exercise Physiology Only	Nutrition Only	Novice (<3)	Experienced (3–6)	Veteran (>6)
**Functions**						
Time spent	11–15 min	11–15 min	16–20 min	11–15 min	11–15 min	16–20 min
Depth †	Most	Moderate	Moderate	Moderate	Most	Moderate
Importance ‡	Most	Most	Most	Moderate	Most	Most
**Measuring water balance and calculating fluid loss**						
Time spent	16–20 min	11–15 min	16–20 min	16–20 min	16–20 min	16–20 min
Depth	Most	Moderate	Most	Most	Most	Most
Importance	Moderate	Most	Most	Most	Most	Most
**Over/under hydration and fluid related problems**						
Time spent	16–20 min	16–20 min	16–20 min	>20 min	16–20 min	16–20 min
Depth	Most	Most	Most	Most	Most	Most
Importance	Most	Most	Most	Most	Most	Most
**Timing (before, during, after exercise)**						
Time spent	16–20 min	16–20 min	16–20 min	>20 min	16–20 min	16–20 min
Depth	Most	Most	Most	Most	Most	Most
Importance	Most	Most	Most	Most	Most	Most
**Assessing fluid needs**						
Time spent	16–20 min	16–20 min	16–20 min	>20 min	11–15 min	16–20 min
Depth	Most	Most	Most	Most	Most	Most
Importance	Most	Most	Most	Most	Most	Most
**Environmental factors**						
Time spent	16–20 min	16–20 min	11–15 min	16–20 min	11–15 min	16–20 min
Depth	Moderate	Most	Most	Most	Most	Most
Importance	Most	Most	Most	Most	Most	Most
**Absorption**						
Time spent	11–15 min	11–15 min	11–15 min	16–20 min	11–15 min	11–15 min
Depth	Moderate	Moderate	Moderate	Most	Moderate	Moderate
Importance	Moderate	Moderate	Most	Most	Most	Moderate

† Categories of Depth scores based on a scale of 1–10. Least: 1–3.0. Moderate: 3.1–6.9. Most: 7.0–10. ‡ Categories of Importance scores based on a scale of 1–10. Least: 1–3.0. Moderate: 3.1–6.9. Most: 7.0–10

**Table 4 sports-11-00176-t004:** Comparison of the average time spent, depth and importance of the major topics for **eating disorders** reported by sports nutrition instructors in a given semester based on the instructor’s degree, age, and experience teaching the course (N = 41).

	Instructor Educational Preparation			Number of Times Course Taught		
Topic	Both Nutrition and Exercise Physiology	Exercise Physiology Only	Nutrition Only	Novice (<3)	Experienced (3–6)	Veteran (>6)
**Types of eating disorders**						
Time spent	6–10 min	1–5 min	11–15 min	6–10 min	6–10 min	6–10 min
Depth †	Moderate	Moderate	Moderate	Moderate	Moderate	Moderate
Importance ‡	Moderate	Moderate	Most	Moderate	Moderate	Moderate
**Orthorexia**						
Time spent	6–10 min	1–5 min	6–10 min	6–10 min	1–5 min	1–5 min
Depth	Moderate	Moderate	Moderate	Moderate	Moderate	Moderate
Importance	Moderate	Moderate	Moderate	Moderate	Moderate	Moderate
**Female athlete triad**						
Time spent	11–15 min	11–15 min	11–15 min	16–20 min	11–15 min	11–15 min
Depth	Moderate	Most	Most	Most	Most	Moderate
Importance	Moderate	Most	Most	Most	Most	Most
**Disordered eating**						
Time spent	11–15 min	6–10 min	11–15 min	11–15 min	6–10 min	11–15 min
Depth	Moderate	Moderate	Most	Most	Most	Most
Importance	Most	Moderate	Most	Most	Most	Most
**Signs and symptoms**						
Time spent	11–15 min	6–10 min	11–15 min	11–15 min	6–10 min	11–15 min
Depth	Moderate	Moderate	Most	Most	Moderate	Moderate
Importance	Moderate	Moderate	Most	Most	Most	Moderate
**Causes**						
Time spent	6–10 min	6–10 min	11–15 min	16–20 min	6–10 min	6–10 min
Depth	Moderate	Moderate	Moderate	Most	Moderate	Moderate
Importance	Moderate	Most	Most	Most	Most	Moderate
**Treatments**						
Time spent	6–10 min	6–10 min	6–10 min	11–15 min	6–10 min	6–10 min
Depth	Moderate	Moderate	Moderate	Moderate	Moderate	Moderate
Importance	Moderate	Moderate	Moderate	Most	Most	Moderate
**Hormonal adaptations**						
Time spent	6–10 min	6–10 min	6–10 min	16–20 min ^a^	1–5 min ^b^	6–10 min ^b^
Depth	Moderate	Moderate	Moderate	Most	Moderate	Moderate
Importance	Moderate	Moderate	Moderate	Most	Moderate	Moderate

† Categories of Depth scores based on a scale of 1–10. Least: 1–3.0. Moderate: 3.1–6.9. Most: 7.0–10. ‡ Categories of Importance scores based on a scale of 1–10. Least: 1–3.0. Moderate: 3.1–6.9. Most: 7.0–10. ^a,b^ Means within categories (degree type, age, experience) on each row are significantly different from each other; *p* < 0.001.

## Data Availability

Data will be made available by the corresponding author upon request.

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
