# Peer review of "Educational Preparation and Course Approach of Undergraduate Sports Nutrition instructors in Large U.S. Institutions"

_sports, 2023, doi:10.3390/sports11090176_

Round 1

Reviewer 1 Report

General

The current manuscript looks to evaluate the education background and time dedicated to course content in undergraduate Sports Nutrition courses at large U.S. institutions. The study revealed that there was little difference between course content based on education backgrounds and that time dedicated towards specific course content was similar among all participants. While this study is interesting and has some potential value, I believe there are significant revisions that need to be addressed before this manuscript can be published.

COMMENTS

General

The introduction lacks specific content on what makes this manuscript important. A stronger background needs to be provided to support your purpose. We are making the assumption that students who take these courses, specifically in exercise physiology or nutrition departments are using this information as they move forward. Some background on what graduates of these programs typically do post-graduation would be beneficial.

Introduction

Page 2, Line 50: Remove the word ‘in’

Methods

In general, your methods do not have enough content if someone was interested in replication your study. Details need to be provided on specific questions asked during the survey, along with rating criteria. This may be easier as an appendix or can be included in the document. If done in the document itself, be sure it follows the same format from the results and discussion.

Page 2, Lines 85-88: How did you get email addresses for potential participants? You only state how you got a list of universities.

Page 3, Lines 136-139: How do you know this information about participants who did not participate? Was this part of your methods, as well as your IRB, that you’d be collecting this information? I would not report this information as it doesn’t add anything to the manuscript.

Results

Table 1 & 2: Under EPO, Functions, you have 16-10 mins on both tables. Please correct these typos.

Discussion

Your discussion is well formatted and has some good points, but seems like you are restating your results again. I think you’ll have a stronger discussion once you improve the introduction. Why is this important in the aspect of how much time is spent on this in each course?

Author Response

Thank you for the feedback.  It help strengthen our manuscript.

Responses to Reviewer 1

General 

The introduction lacks specific content on what makes this manuscript important. A stronger background needs to be provided to support your purpose. We are making the assumption that students who take these courses, specifically in exercise physiology or nutrition departments are using this information as they move forward. Some background on what graduates of these programs typically do post-graduation would be beneficial.

 This was a helpful comment.  We have added several sentences to the second paragraph in the introductory section to strengthen the importance of the study.  Thank you

Page 2, Line 50: Remove the word ‘in’.  Completed. 

 Methods

In general, your methods do not have enough content if someone was interested in replication your study. Details need to be provided on specific questions asked during the survey, along with rating criteria. This may be easier as an appendix or can be included in the document. If done in the document itself, be sure it follows the same format from the results and discussion.

Under section 2.1 in the methods section, we indicated that the rating scale for the questions was 1-10.  We also indicated that the topics are found in the tables.  However, we agree that adding a copy of the survey as an appendix would be helpful for someone who wants to replicate the study.

Page 2, Lines 85-88: How did you get email addresses for potential participants? You only state how you got a list of universities.

In section 2.2 of the methods section we added the following:

A total of 324 institutions were identified.  Research assistants then went to each university’s website to identify if a sports nutrition course was offered.  There were 189 universities that offered a sports nutrition course, with 217 instructors.  Emails were identified from the websites of each institution.  

Page 3, Lines 136-139: How do you know this information about participants who did not participate? Was this part of your methods, as well as your IRB, that you’d be collecting this information? I would not report this information as it doesn’t add anything to the manuscript.

This section was included in an effort to show that our sample was likely representative of the broader population.  We believe it is important for the manuscript, but would consider removing it if necessary.

 Results

Table 1 & 2: Under EPO, Functions, you have 16-10 mins on both tables. Please correct these typos.  These corrections were completed

Reviewer 2 Report

What are the practical applications of the study? What is the purpose of publishing this information? How do the authors believe that this information can help other investigators? Do the aspects discussed do not refer to US standards? What is the impact of this study for science?

Author Response

Thank you for your feedback.  It strengthened our manuscript.

 Responses to Reviewer 2

Your discussion is well formatted and has some good points, but seems like you are restating your results again. I think you’ll have a stronger discussion once you improve the introduction. Why is this important in the aspect of how much time is spent on this in each course?

We agree that the discussion section seems similar to the results.  However, they are different.  In the discussion section we elaborate on findings for the theoretical verses the applied principles, which was one of our main objectives.  We also summarize some differences between the two instructor backgrounds (EPD and NOD) for data we could not include in the results.  We had many tables (22 supplemental tables) the showed specific differences between the two instructor backgrounds to include in this manuscript, so we elected to highlight a few helpful results from the “data not shown”.  Finally, we include some peer-reviewed referenced discussion on why different topics were important and needed for the course.

This is a paper evaluating teaching in nutrition in US universities.

It provides an interesting perspective and an in deep analysis which can be useful for designing the nutrition course. 

However, the results are a little surprising, because the Authors didn't find any significant differences in teaching among the different courses. The reason must be further discussed. Maybe because the nutrition science is a very narrow field and all teachers use the same sources for information ? 

We agree completely that not finding significant differences in how this course was taught by instructors of different background was surprising.  We expected to see differences, especially in time spent on applied vs theoretical topics.  We also agree with your initial comment that it might be because the textbooks are so similar and rely on similar evidence and evidence based guidelines.  We added potential interpretations of our findings at the end of the first paragraph of the discussion section.   Thank you for this suggestion.  It strengthens the paper.

It also surprising that any attention was given to biochemistry and chemistry aspects, that should be linked with nutrition. The Authors have intentionally avoided to asking about this important part of teaching ?

This is a good point.  When we designed the study, we tried to streamline the topics in the survey, so we considered chemistry and biochemistry under the heading “metabolism”.  Justification for this decision was the likelihood that the course might be taught and a 100 or 200 level, so there would likely be minimal chemistry or biochemistry, unless they were prereqs.  After the data were collected it was revealed that the most common pre-reqs were human nutrition and human physiology. 

Reviewer 3 Report

This is a paper evaluating teaching in nutrition in US universities.

It provides an interesting perspective and an in deep analysis which can be useful for designing the nutrition course.

However, the results are a little surprising, because the Authors didn't find any significant differences in teaching among the different courses. The reason must be further discussed. Maybe because the nutrition science is a very narrow field and all teachers use the same sources for information ? 

It also surprising that any attention was given to biochemistry and chemistry aspects, that should be linked with nutrition. The Authors have intentionally avoided to asking about this important part of teaching ?

Author Response

However, the results are a little surprising, because the Authors didn't find any significant differences in teaching among the different courses. The reason must be further discussed. Maybe because the nutrition science is a very narrow field and all teachers use the same sources for information ? 

We agree completely that not finding significant differences in how this course was taught by instructors of different background was surprising.  We expected to see differences, especially in time spent on applied vs theoretical topics.  We also agree with your initial comment that it might be because the textbooks are so similar and rely on similar evidence and evidence based guidelines.  We added potential interpretations of our findings at the end of the first paragraph of the discussion section.   Thank you for this suggestion.  It strengthens the paper.

It also surprising that any attention was given to biochemistry and chemistry aspects, that should be linked with nutrition. The Authors have intentionally avoided to asking about this important part of teaching ?

This is a good point.  When we designed the study, we tried to streamline the topics in the survey, so we considered chemistry and biochemistry under the heading “metabolism”.  Justification for this decision was the likelihood that the course might be taught and a 100 or 200 level, so there would likely be minimal chemistry or biochemistry, unless they were prereqs.  After the data were collected it was revealed that the most common pre-reqs were human nutrition and human physiology. 

Reviewer 4 Report

Thank you very much for the opportunity to review this manuscript. The topic is interesting, however, I would like to indicate numerous comments about this study.

Please provide the main statistical findings in the abstract.

The introduction provides limited information about the current state of knowledge regarding this topic and the study rationale. Please elaborate on that.

Is nutrition a regulated course at US universities? How flexible are the national requirements for the teachers to adjust their courses according to their preferences? Was it impacted by the textbook's content?

Line 74 – Please provide the survey as supplementary material.

Line 90 – Please provide the approval number.

The survey among the teachers cannot provide reliable information about the feedback., it would be helpful to compare this information with the students' feedback, especially regarding the practical aspects of the course.

Was the ZOOM interview conducted among all the participants or only selected – if so, how were they selected?

41 participants is a relatively low number of respondents regarding the observational type of the study. I suggest extending the observation – (recruiting additional participants).

A cross-sectional correlation does not provide the reason-effect relationship, so please do not claim the impact of selected factors. You should assess this aspect prospectively to admit whether particular aspects impact the content of the courses.

Please elaborate on the rationale why selected aspects of the courses were analyzed, why are these particular aspects crucial in terms of nutritional education?

There is no information about the statistical analysis results, what are the p-values?

Extensive English language revision is also advised.

Extensive English language revision is also advised.

Author Response

Please provide the main statistical findings in the abstract.

There were very few significant differences to report, and those few differences were of minor importance, and not likely practical.  So, while we agree that an abstract should report statistical numbers, including p-values, sometimes descriptive studies do not lend themselves to typical statistical findings.   But we believe the language in the abstract describes the major findings as such:

“The findings indicated that regardless of the instructor’s educational preparation and discipline, the majority of sports nutrition topics received similar time and depth and were rated as similarly important.  Out of 10 current textbooks, the majority of instructors preferred only one of four of them.  From the short interviews, instructors reported that their courses were more applied than theoretical or balanced between the two.  Most instructors designed their courses with a focus on achieving applied outcomes.”

The introduction provides limited information about the current state of knowledge regarding this topic and the study rationale. Please elaborate on that.

 This was a helpful comment.  We have added several sentences to the second paragraph and included below in the introductory section to strengthen the importance of the study.  Thank you

“These students are most likely to work with athletes in their professional careers such as sports nutritionists, sports dietitians, athletic trainers, exercise physiologists, or sports medicine doctors and it is important that these students are adequately taught the correct scientific foundational knowledge and skills to work with athletes. They might work with all classes of athletes, from amateurs to Olympians or professionals. Knowing and teaching sound, evidenced-based sports nutrition principles will help maximize athletic performance while maintaining the athlete’s emotional and physical well-being. Additionally, other students might take the course for their own personal interests who also need to be taught evidenced-based nutritional concepts to support their physical activity while also maintaining emotional and physical well-being”.

Is nutrition a regulated course at US universities? How flexible are the national requirements for the teachers to adjust their courses according to their preferences? Was it impacted by the textbook's content?

This is a good question. It isn’t likely that any nutrition courses are not regulated; however some professional accrediting bodies (Academy of Nutrition and Dietetics, National Athletic Trainer’s Association)  require that certain general content be covered in a sports nutrition course, but specific details are not mandated.  These two organizations are the most likely organizations to require a sports nutrition course.

Line 74 – Please provide the survey as supplementary material.

It has been included as an appendix

Line 90 – Please provide the approval number. 

The IRB approval number is IRB2021-296.  It has been added to the manuscript.

The survey among the teachers cannot provide reliable information about the feedback., it would be helpful to compare this information with the students' feedback, especially regarding the practical aspects of the course.

The objective of the study was to compare instructor preparation and course approach.  We believe that the survey provided answers to our objectives.  While it would be interesting to obtain student feedback about a course, that would require a different study design as well as IRB approval from all participating institutions.  Perhaps that would be a good next step.  Thank you for the comment.

Was the ZOOM interview conducted among all the participants or only selected – if so, how were they selected?

In section 2.3 of the methods section, we state that the last question of the survey invited subjects to participate in a one on one Zoom follow-up interview.  Participation was voluntary.

41 participants is a relatively low number of respondents regarding the observational type of the study. I suggest extending the observation – (recruiting additional participants).

It would have been nice to have more participants; however, with the sample size we analyzed the results were very consistent between the two groups with no to minimal differences.  An analysis of all the eligible universities identified for this study (nonrespondent, excluded responses and included subjects) over 85% of sports nutrition courses at large US Universities are taught by instructors with degrees in nutrition and or exercise physiology." After consultation with a statistician, it was decided that a larger sample would not likely have changed the results.  Furthermore, we are comfortable that our sample was representative of the larger population.

A cross-sectional correlation does not provide the reason-effect relationship, so please do not claim the impact of selected factors. You should assess this aspect prospectively to admit whether particular aspects impact the content of the courses.

We agree!  This was an observational descriptive study.  We were careful to not make any cause-effect statements.  Therefore, if you noticed any specific cause-effect statements in our manuscript, please let us know where your specific concerns are and we will correct them.

Please elaborate on the rationale why selected aspects of the courses were analyzed, why are these particular aspects crucial in terms of nutritional education?

In section 2.1 of the methods section, we describe how the questions on course content were developed.  We reviewed 10 sports nutrition textbooks (which was all we could find that were published within the last 10 years) to develop questions for the course content.

There is no information about the statistical analysis results, what are the p-values?

The following sentence is found in the methods sections under data analysis: “To reduce the risk of a Type 1 error and to account for multiple comparisons, statistical significance was set at an alpha of p<0.001”.  There were very few significant differences, and they are all indicted with superscripts in the tables.  Because there were so few differences, there is not much to report.  Furthermore, this also mandated that we spend more time focusing on the similarities.

Extensive English language revision is also advised.

The authors of this manuscript are all native English speakers.

Round 2

Reviewer 1 Report

Thank you for addressing my comments. I believe the manuscript is now acceptable for publication.

Author Response

Thank you

Reviewer 2 Report

I accept this article in this form.

Author Response

Thank you

Reviewer 4 Report

Thank you very much for the effort put into the manuscript corrections. I believe that its quality improved.  The Authors' replies are satisfactory, however, there is still one issue I would like to mention.

Line 24 and the results section – Although the differences were not statistically significant, you should provide the p-value data to support this claim.

Author Response

We agree.  The abstract now has 2 additions:

College courses are often offered from various disciplines, and depending on which department offers the class, the course could be taught by faculty with different educational preparation or training.  This could result in significant differences in the approach and content of the course (i.e., theoretical or applied) or difference in the instructors’ perceived importance and therefore, depth and time spent on various topics.   We evaluated potential differences in sports nutrition curriculum because it is a course that is usually taught by either nutritionists or exercise physiologists. A cross-sectional survey was sent to sports nutrition instructors at accredited large U.S. institutions.  Descriptive statistics were analyzed by ANOVA and Χ2 using Crosstabs in Qualtrics. Alpha was set at p<0.001. Additionally, short interviews of some participants were recorded and transcribed verbatim. The findings indicated that regardless of the instructor’s educational preparation and discipline, the majority of sports nutrition topics received similar time and depth and were rated as similarly important (P>0.001).  Out of 10 current textbooks, the majority of instructors preferred only one of four of them.  From the short interviews, instructors reported that their courses were more applied than theoretical or balanced between the two.  Most instructors designed their courses with a focus on achieving applied outcomes.